# Cortico-Subcortical White Matter Bundle Changes in Cervical Dystonia and Blepharospasm

**DOI:** 10.3390/biomedicines11030753

**Published:** 2023-03-01

**Authors:** Costanza Giannì, Claudia Piervincenzi, Daniele Belvisi, Silvia Tommasin, Maria Ilenia De Bartolo, Gina Ferrazzano, Nikolaos Petsas, Giorgio Leodori, Nicoletta Fantoni, Antonella Conte, Alfredo Berardelli, Patrizia Pantano

**Affiliations:** 1Dipartimento di Neuroscienze Umane, Sapienza Università di Roma, 00185 Rome, Italy; 2Radiology Department, IRCCS Neuromed, 86077 Isernia, Italy

**Keywords:** focal dystonia, cervical dystonia, blepharospasm, diffusion tensor imaging, structural MRI, probabilistic tractography, white matter, basal ganglia-thalamo-cortical circuit

## Abstract

Dystonia is thought to be a network disorder due to abnormalities in the basal ganglia-thalamo-cortical circuit. We aimed to investigate the white matter (WM) microstructural damage of bundles connecting pre-defined subcortical and cortical regions in cervical dystonia (CD) and blepharospasm (BSP). Thirty-five patients (17 with CD and 18 with BSP) and 17 healthy subjects underwent MRI, including diffusion tensor imaging (DTI). Probabilistic tractography (BedpostX) was performed to reconstruct WM tracts connecting the globus pallidus, putamen and thalamus with the primary motor, primary sensory and supplementary motor cortices. WM tract integrity was evaluated by deriving their DTI metrics. Significant differences in mean, radial and axial diffusivity between CD and HS and between BSP and HS were found in the majority of the reconstructed WM tracts, while no differences were found between the two groups of patients. The observation of abnormalities in DTI metrics of specific WM tracts suggests a diffuse and extensive loss of WM integrity as a common feature of CD and BSP, aligning with the increasing evidence of microstructural damage of several brain regions belonging to specific circuits, such as the basal ganglia-thalamo-cortical circuit, which likely reflects a common pathophysiological mechanism of focal dystonia.

## 1. Introduction

Dystonia is a movement disorder characterized by abnormal postures and involuntary movements due to repetitive or sustained muscle contractions. It is now thought that dystonia arises through the involvement of a network including the basal ganglia, cerebellum, thalamus and sensorimotor cortices [1,2,3]. In line with this hypothesis, several studies have found abnormalities in the basal ganglia thalamo-cortical circuit in patients with dystonia [1,4,5,6,7,8].

In patients with the two most frequent forms of focal/segmental dystonia, characterized by clinical involvement of a single body part, namely cervical dystonia (CD) and blepharospasm (BSP), diffusion tensor imaging (DTI) studies demonstrated WM changes [9] in several structures including the basal ganglia and cerebellum [10,11,12]. Microstructural alterations were also found in the white matter (WM) adjacent to the primary sensorimotor, inferior parietal and middle cingulate cortices in patients with CD [13,14,15] and BSP [16,17,18]. Finally, studies with whole-brain approaches showed WM microstructural disruption in the corpus callosum, the internal capsule and the white matter underlying the sensorimotor cortex in CD and BSP patients [11,19,20].

In CD, tractography-based studies also demonstrated abnormal connections between infratentorial structures and the basal ganglia; specifically, between the pallidum and brainstem [21], between the thalamus, middle frontal gyrus and brainstem [22], and within the dentato-rubro-thalamic tract [23]. However, there have been no studies conducting tractography in BSP, and it is unknown whether CD and BSP have specific microstructural abnormalities, in line with recently demonstrated functional alterations [4], or whether they share similar abnormalities to the basal ganglia thalamo-cortical network.

In this paper, we investigate in CD and BSP the possible microstructural changes of WM bundles connecting predefined subcortical and cortical regions involved in the network underlying the pathophysiology of focal dystonia. Using a probabilistic tractography approach [24], we reconstruct WM tracts connecting the globus pallidus, putamen, and thalamus with primary motor, primary sensory, and supplementary motor cortices. We then evaluate the integrity of those WM tracts by deriving their DTI metrics. Finally, we investigate possible correlations between WM microstructural damage and clinical features of dystonic patients.

## 2. Materials and Methods

### 2.1. Participants and Clinical Assessment

Patients were consecutively recruited from the movement disorder outpatient clinic of the Department of Human Neurosciences, Sapienza University of Rome (Italy). Patient inclusion criteria consisted of a clinical diagnosis of CD or BSP according to diagnostic criteria [25] and age > 18 years old. Exclusion criteria were neurological abnormalities other than tremor, psychiatric diseases, concomitant systemic disease (e.g., diabetes, liver disease, chronic renal failure, cardiovascular diseases) or contraindications to MRI. Forty-two patients with adult-onset focal dystonia were enrolled. The Toronto Western Spasmodic Torticollis Rating Scale (TWSTRS) [26] and the Blepharospasm Severity Rating Scale (BSRS) [27] were used to assess the severity of CD and BSP, respectively, while quality of life and disability were evaluated using the Cervical Dystonia Impact Profile (CDIP-58) for CD and the Blepharospasm Disability Index (BSDI) for BSP. Disease duration and handedness were recorded for all patients. All patients were evaluated at least 3 months after the last botulinum toxin injection to exclude any possible confounders due to the botulinum neurotoxin effect. None of the patients were under other treatment. A group of 17 age- and sex-matched healthy subjects (HS) from a pool of volunteers was enrolled as a control group. All the participants gave their informed consent and the experimental procedure was approved by the ethics committee of Sapienza University of Rome (CE n 4041, 24 March 2016) and conducted in accordance with the Declaration of Helsinki.

### 2.2. MRI Acquisition

All participants underwent a multimodal 3T MRI scan (12-channel head coil for parallel imaging, Verio, Siemens AG), including (1) high-resolution 3-dimensional T1-weighted MPRAGE (3D T1) with 176 contiguous sagittal slices, 1 mm thick slice (repetition time (TR) = 1900 ms, echo time (TE) = 2.93 ms, flip angle = 9°, matrix = 256 × 256, field of view (FOV) = 260 mm^2^; (2) T2-weighted images (TR = 3320 ms, TE = 10/103 ms, FOV = 220 mm^2^, 384 × 384 matrix, 4 mm thick slices, 30% gap); and (3) diffusion-tensor imaging (DTI), single-shot echo-planar spin-echo sequence, with one b = 0 and 30 gradient directions, b = 0 and 1000 s/mm^2^, TR = 12,200 ms, TE = 94 ms, FOV = 192 mm, matrix = 96 × 96, 72 axial 2 mm thick slices, no gap.

### 2.3. MRI Data Analysis

Before data analysis, all images were visually inspected for a qualitative assessment of artifacts.

Structural preprocessing was performed with FMRIB’s Software Library (FSL), version 6.0.1 (https://fsl.fmrib.ox.ac.uk/fsl (accessed on 1 September 2022)). Diffusion data were visually inspected for artifacts and preprocessed using different tools from FDT (FMRIB Diffusion Toolbox, part of FSL (FMRIB’s Software Library v.6.0.4, http://www.fmrib.ox.ac.uk/fsl/) (accessed on 15 September 2022). Images were corrected for eddy current distortion and head motion using a 12-parameter affine registration to each subject’s first no-diffusion weighted volume, and the gradient directions were rotated accordingly [28]. Non-brain tissue was removed from the eddy-corrected images using the Brain Extraction Tool, part of FSL (FMRIB’s Software Library v.6.0.4, http://www.fmrib.ox.ac.uk/fsl/) (accessed on 15 September 2022) (BET; [29]), creating a binary mask of the brain. Then, maps of fractional anisotropy (FA), mean diffusivity (MD), axial diffusivity (AD) and radial diffusivity (RD) were estimated at the individual level using the DTIFIT tool, part of FSL (FMRIB’s Software Library v.6.0.4, http://www.fmrib.ox.ac.uk/fsl/) (accessed on 30 September 2022) by fitting a tensor model to the eddy-corrected and brain masked diffusion data. Registration between diffusion, structural and standard space images was performed within FDT. Transformation matrices, and their inverses, were created to transform images between spaces. 

#### 2.3.1. Selection of ROIs

To reconstruct WM tracts between the sensorimotor cortex and subcortical structures, regions of interest (ROIs) were defined. We use probabilistic atlases to identify cortical regions of interest (ROIs): the primary motor cortex (M1) (head/face region) and the primary somatosensory cortex (S1) (face/upper limb region) were derived from the Brainnetome atlas (https://atlas.brainnetome.org/download.html (accessed on 15 September 2022)) [30]), and the juxtapositional lobule cortex (formerly supplementary motor cortex—SMA) was identified from the Harvard-Oxford Cortical Structural Atlas (http://www.fmrib.ox.ac.uk/fsl/data/atlas descriptions.html (accessed on 15 September 2022)). We thresholded at 25% M1, S1 and SMA ROIs and then divided on the sagittal plane x = 0 in the right and left regions. We used FIRST-FSL to identify subcortical ROIs in each patient: left and right globi pallidi, putamen and thalami. Finally, we registered cortical regions from standard space and subcortical regions from structural space into subject diffusion space and visually checked for accuracy.

#### 2.3.2. Tractography

We performed probabilistic tractography within each participant’s diffusion space using BedpostX, part of FSL (FMRIB’s Software Library v.6.0.4, http://www.fmrib.ox.ac.uk/fsl/) (accessed on 1 October 2022) [31] with default parameters. We generated streamline probability distribution maps between each predefined subcortical and cortical region of interest (ROI). In each reconstructed map, we specified the subcortical region as the seed, the cortical region as the target and the contralateral hemisphere as the exclusion mask. We also specified the cortical target region as a termination mask, to identify the only and exact connections between the given seed and the given target [32]. We then normalized pathway probability maps for seed size by dividing the probability maps by the total number of successfully generated streamlines, and we removed spurious connections by thresholding the resulting maps by 5% [32,33]. We then binarized thresholded probability maps and overlaid of FA, MD, AD and RD on individual maps, from which we extracted average values [34] to evaluate WM tracts’ integrity.

### 2.4. Statistical Analysis

Statistical analysis was performed using SPSS software (IBM SPSS Statistics, version 25.0, IBM Corp., Armonk, NY, USA). One-way ANOVA was used to compare age and the χ2 test was used to compare sex between patients and HS. Group differences in terms of DTI (FA, MD, RD and AD) measures within the WM tracts of interest were tested via multivariate analysis (Kruskal–Wallis). The significance level was set at *p* < 0.05, Bonferroni-corrected for multiple comparisons.

To correlate WM microstructural damage of the tracts of interest with clinical scales, altered DTI metrics of each WM bundle were non-parametrically correlated via Spearman’s correlation test (Bonferroni corrected for multiple comparisons) with TWSTRS and CDIP-58 for the CD patients and with the BSRS and BSDI for the BSP group. Subsequently, to limit the number of correlations, we derived indexes of global damage of subcortical-sensorimotor cortices WM tracts for each patient by averaging FA, MD, AD and RD values of all reconstructed WM bundles, thus obtaining the FA index, MD index, AD index and RD index. Each index was then correlated with clinical scores via a non-parametric test (Spearman’s correlation test). The significance level was set at *p* < 0.05, Bonferroni-corrected for multiple comparisons.

## 3. Results 

Forty-two patients with adult-onset focal dystonia and 17 HS were enrolled in the study. Due to motion artifacts in MRI images, seven patients were excluded (five with CD and two with BSP). Thirty-five patients (17 with CD and 18 with BSP) and 17 HS were included in the analysis. No differences in age (F = 2.35, *p* = 0.09) or sex (F = 2.89, *p* = 0.06) were found between the three groups. All subjects were right-handed. The demographic and clinical characteristics of study participants are reported in Table 1.

Streamlines of WM tracts were successfully generated for all participants (Figure 1). Significant between-group differences in MD, RD and AD were found in the majority of the reconstructed WM tracts, while FA was significantly different in one WM tract alone (Table 2, Table 3, Table 4 and Table 5). Post hoc testing (Dunn–Bonferroni) showed significant differences between CD and HS and between BSP and HS, while no differences were found between the two groups of patients. Specifically, patients showed lower FA and higher MD, RD and AD compared to HS (Table 2, Table 3, Table 4 and Table 5 and Figure 2, Figure 3, Figure 4 and Figure 5).

In patients with BSP, a significant positive correlation was found between BSRS and the MD and RD of all WM bundles (data not shown). No correlation was found between the extent of WM damage and either TWSTRS or CDIP-58 in the CD group. When correlating indexes of global damage of subcortical-sensorimotor cortices WM tracts with clinical scales, a significant positive correlation was found between the MD and RD indexes and BSRS in patients with BSP (Figure 6).

## 4. Discussion

This study investigated white matter microstructural features in the two most frequent types of adult-onset focal dystonia, CD and BSP. For the first time, we studied CD and BSP patients with a methodology recently used for WM tract reconstruction in patients with embouchure dystonia [24] to evaluate specific WM bundles with a probabilistic tractography approach. We found that both forms of dystonia shared extensive microstructural changes of WM bundles of the basal ganglia-sensorimotor network, without any DTI parameter able to differentiate one form of dystonia from the other. The analysis of specific WM bundles connecting subcortical structures and sensorimotor cortices showed extensive fiber loss in CD and BSP compared to HS, with no differences between the two groups of patients. 

Only a few tractography-based studies demonstrated abnormalities in WM tracts in patients with CD while focusing on tracts between infratentorial structures and basal-ganglia, specifically, between the pallidum and brainstem [21], between the thalamus, middle frontal gyrus and brainstem [22], and within the dentato-rubro-thalamic tract [23].

In the present study, subcortical ROIs coincided with the putamen and the globus pallidus as the primary basal ganglia input and output structures, respectively, and the thalamus as a relay structure between the basal ganglia, cerebellum and cortex. Cortical ROIs were identified in the head/face regions of the primary sensory and motor cortices and the SMA. The choice of investigating direct cortico-pallidal connectivity was based on animal studies describing direct projections from the cerebral cortex to the globus pallidus [35,36] and on recent diffusion tractography studies showing direct cortico-pallidal projections in humans [37,38,39], relevant in the pathophysiology of dystonia [40,41].

In CD and BSP patients, DTI analysis revealed a diffuse increase in MD, RD and AD in the majority of the reconstructed WM tracts, and an FA reduction limited to pallidum–SMA, without differences between the two groups of patients. An increase in MD, which reflects cellular density and extracellular volume [42,43], indicates a less organized myelin and/or axonal structure [44], while increased AD and RD, which give information about the spatial orientation of fibers, suggest prevalent axonal damage [45] and demyelination [46], respectively. Reduced FA can be caused by the degradation of myelin sheaths and/or axonal membranes [45,47,48]. Overall, data of the present study support the hypothesis of axonal and myelin loss due to microstructural abnormalities of the basal ganglia-thalamo-cortical circuit, with alterations of both direct and indirect pathways [49] and direct cortico-pallidal pathways [37,40,50]. 

The results of the present study showing changes in specific white matter tracts expand previous literature on CD and BSP, demonstrating diffuse microstructural damage in the basal ganglia, cerebellum and sensorimotor cortical areas [10,11,12,13], as well as in the white matter [11,19,20,51,52]. Moreover, the WM tracts we reconstructed correspond to brain regions with microstructural integrity loss in the WM adjacent to the pallidum and putamen and the precentral and postcentral gyri [13,17,53]. Unlike our results, Berman and colleagues found different patterns of altered microstructural WM changes in CD and BSP. Specifically, when comparing CD and BSP patients, reduced FA in the cerebellum and the bilateral caudate nucleus was found in CD patients, whereas reduced FA in the globus pallidum internus and the red nucleus was found in BSP patients [10]. The reasons for our different findings are probably the different methodological approaches of the studies and the different brain regions investigated.

We also found a significant correlation between the MD and RD of all reconstructed WM tracts and the severity of blepharospasm. This finding is in line with previous studies that showed a correlation between altered DTI metrics in subcortical structures [10,12] and long WM tracts [54] with clinical scales in BSP. The absence of a correlation between the extent of WM damage and clinical scales in patients with CD is also consistent with previous studies [10,23,52]. The development of unbiased and reliable clinical scales for focal dystonia is an important field of research in the current literature.

This study is not without limitations. The cross-sectional design makes it impossible to conclude whether the changes we described are causative or compensatory. DTI allows non-invasive in vivo assessment of brain structural connectivity; however, caution is needed when interpreting the results given the intrinsic limitations of the DTI technique for defining the direction of structural connection change. The availability of more objective and reliable clinical scales not biased by patient perception could overcome the difficulty of making clinical–radiological correlations. 

To conclude, the present observation of changes in DTI metrics of specific WM tracts suggests a diffuse and extensive alteration in WM integrity as a common feature of two forms of focal dystonia, namely cervical dystonia and blepharospasm. The present results align with the increasing evidence of microstructural damage to several brain WM bundles belonging to a specific circuit, i.e., the basal ganglia-thalamo-cortical circuit. Altered structural connectivity between the basal ganglia and sensorimotor cortices parallels functional connectivity abnormalities consistently reported in the basal ganglia-thalamo-sensorimotor circuit in cervical dystonia and blepharospasm, likely indicating a common pathophysiological mechanism underlying both forms of focal dystonia. 

## Figures and Tables

**Figure 1 biomedicines-11-00753-f001:**
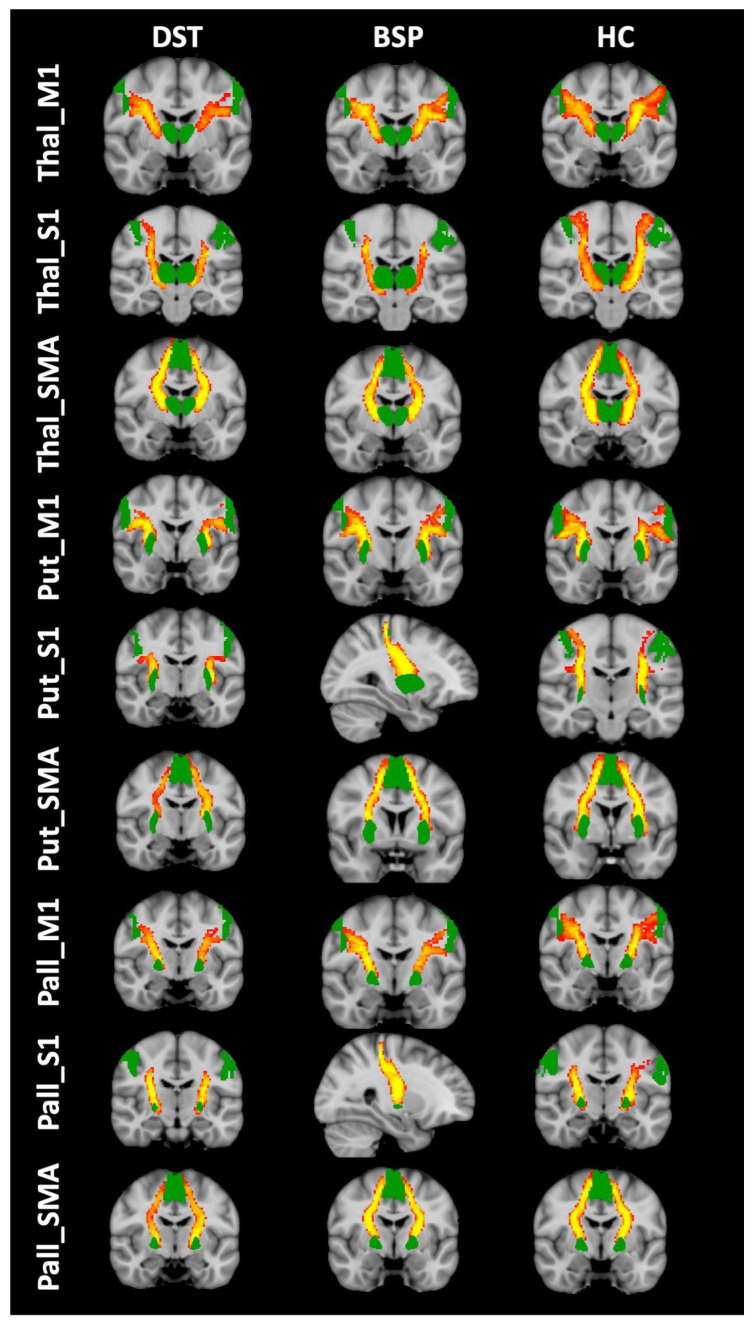
Reconstructed white matter tracts in healthy subjects (HC) ((**right**) DST patients (**left**) and BSP patients (**center**) overlaid on the MNI152 standard brain). Red-yellow colors mean the extent of spatial overlap of reconstructed tracts between participants; specifically, red indicates at least 50% overlap and yellow indicates 100%. Green areas are the regions of interest used for probabilistic tractography.

**Figure 2 biomedicines-11-00753-f002:**
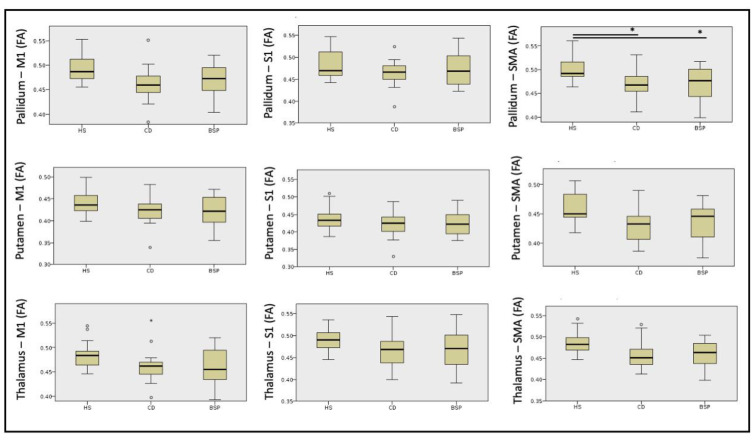
Differences in fractional anisotropy (FA) of cortical-subcortical white matter tracts among subjects. HS: healthy subjects; CD: cervical dystonia; BSP: blepharospasm; M1: primary motor cortex head/face region; S1: primary sensory cortex head/upper limb region; SMA: supplementary motor area. Kruskal–Wallis post hoc analysis (* *p* < 0.05). Circles indicates outliers.

**Figure 3 biomedicines-11-00753-f003:**
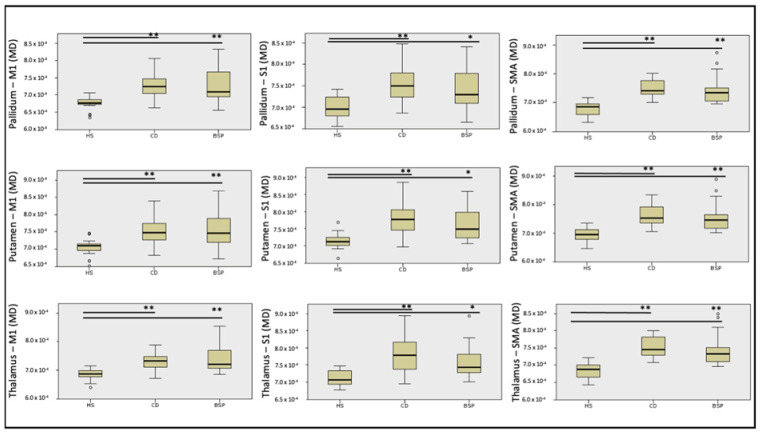
Differences in mean diffusivity (MD) of cortical-subcortical white matter tracts among subjects. HS: healthy subjects; CD: cervical dystonia; BSP: blepharospasm; M1: primary motor cortex head/face region; S1: primary sensory cortex head/upper limb region; SMA: supplementary motor area. Kruskal–Wallis post hoc analysis (* *p* < 0.05, ** *p* < 0.001). Circles indicates outliers.

**Figure 4 biomedicines-11-00753-f004:**
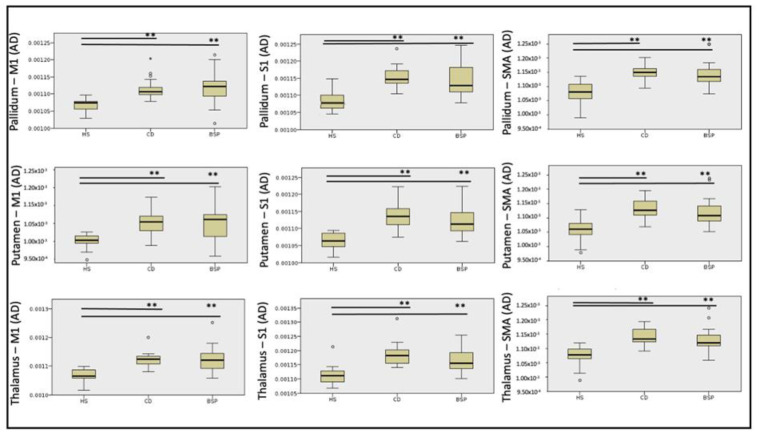
Differences in axial diffusivity (AD) of cortical-subcortical white matter tracts among subjects. HS: healthy subjects; CD: cervical dystonia; BSP: blepharospasm; M1: primary motor cortex head/face region; S1: primary sensory cortex head/upper limb region; SMA: supplementary motor area. Kruskal–Wallis post hoc analysis (** *p* < 0.001). Circles indicates outliers.

**Figure 5 biomedicines-11-00753-f005:**
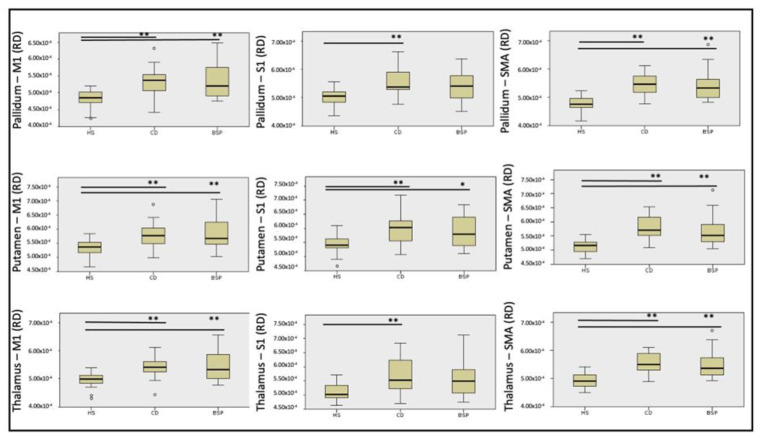
Differences in radial diffusivity (RD) of cortical-subcortical white matter tracts among subjects. HS: healthy subjects; CD: cervical dystonia; BSP: blepharospasm; M1: primary motor cortex head/face region; S1: primary sensory cortex head/upper limb region; SMA: supplementary motor area. Kruskal–Wallis post hoc analysis (* *p* < 0.05, ** *p* < 0.001). Circles indicates outliers.

**Figure 6 biomedicines-11-00753-f006:**
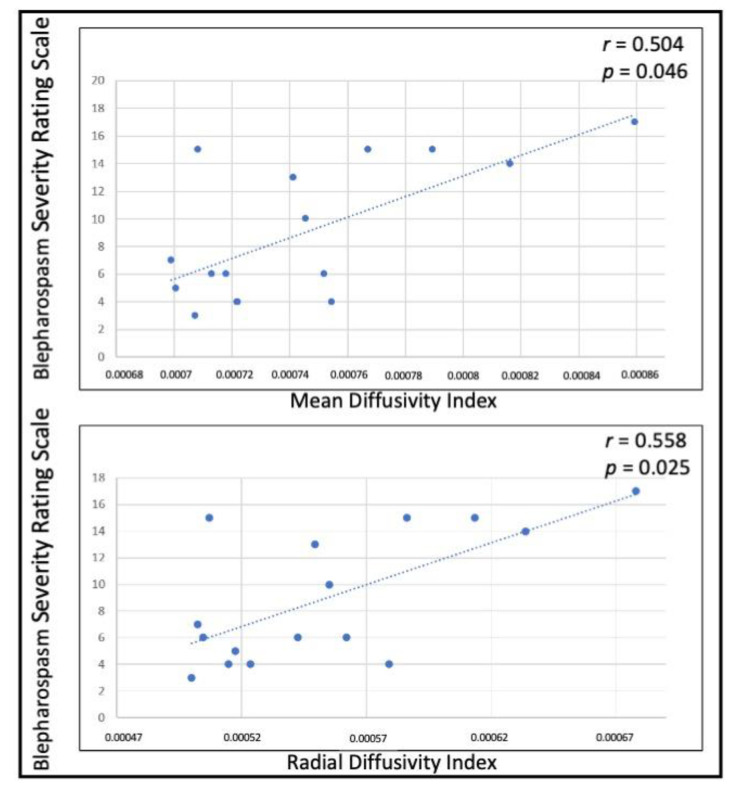
Correlation between diffusion tensor imaging metrics and blepharospasm severity scale is shown. Spearman’s correlation test (*r* and *p* are displayed).

**Table 1 biomedicines-11-00753-t001:** Demographic and clinical characteristics of patients and healthy subjects.

Group (Subjects)	Cervical Dystonia (*n* = 17)	Blepharospasm (*n* = 18)	Healthy Subjects (*n* = 17)	*p*-Value
Age, years	55.7 (10.1)	61.5 (8.8)	54.4 (13.9)	>0.05
Sex (female/male)	14/3	12/6	8/9	>0.05
Disease duration, years	13.9 (9.9)	11.6 (3.8)	-	>0.05
TWSTRS	16.6 (11.2)	-	-	NA
CDIP-58	56.4 (19.1)	-	-	NA
BSRS	-	9.1 (4.7)	-	NA
BSDI	-	9.5 (6.6)	-	NA
Head tremor (Y/N)	9/8	NA	NA	NA

Data are shown as mean (standard deviation). Differences between groups were assessed using the t test; differences for sex were assessed using the χ2 test. NA = not applicable; TWSTRS = Toronto Western Spasmodic Torticollis Rating Scale; CDIP-58 = Cervical Dystonia Impact Profile; BSRS = Blepharospasm Severity Rating Scale; BSDI = Blepharospasm Disability Index; Y = yes; N = no; R = right; L = left.

**Table 2 biomedicines-11-00753-t002:** Fractional anisotropy of subcortical-cortical WM tracts.

	Healthy Subjects (*n* = 17)	Cervical Dystonia (*n* = 17)	Blepharospasm (*n* = 18)	*p*	H	Post Hoc
PALLIDUM—M1 Mean (SD)	0.494 (0.03)	0.462 (0.03)	0.467 (0.03)	>0.05	8.75	-
PALLIDUM—S1 Mean (SD)	0.482 (0.03)	0.463 (0.03)	0.472 (0.03)	>0.05	2.58	-
PALLIDUM—SMA Mean (SD)	0.504 (0.03)	0.468 (0.03)	0.469 (0.03)	**0.005**	10.74	**HS-BSP *p* = 0.026**
						**HS-CD *p* = 0.008**
						BSP-CD ns
PUTAMEN—M1 Mean (SD)	0.443 (0.03)	0.421 (0.03)	0.423 (0.03)	>0.05	3.68	-
PUTAMEN—S1 Mean (SD)	0.441 (0.03)	0.423 (0.03)	0.426 (0.04)	>0.05	1.65	-
PUTAMEN—SMA Mean (SD)	0.459 (0.03)	0.432 (0.03)	0.435 (0.03)	>0.05	5.56	-
THALAMUS—M1 Mean (SD)	0.485 (0.03)	0.462 (0.03)	0.461 (0.04)	>0.05	5.74	-
THALAMUS—S1 Mean (SD)	0.489 (0.03)	0.468 (0.04)	0.472 (0.04)	>0.05	3.76	-
THALAMUS—SMA Mean (SD)	0.487 (0.03)	0.457 (0.03)	0.459 (0.03)	>0.05	9.15	-

Differences between the three groups (HS, CD and BSP) were assessed using the Kruskal–Wallis test (Bonferroni corrected for multiple comparisons). M1 = bilateral primary motor cortex, head/face region; S1 = bilateral primary sensory cortex, head/upper limb region; SMA = supplementary motor area; SD = standard deviation. Significant *p* are shown in bold. ns = not significant.

**Table 3 biomedicines-11-00753-t003:** Mean diffusivity of subcortical-cortical WM tracts.

Mean Diffusivity						
	Healthy Subjects (*n* = 17)	Cervical Dystonia (*n* = 17)	Blepharospasm (*n* = 18)	*p*	H	Post Hoc
PALLIDUM—M1 Mean (SD)	0.00068 (0.00002)	0.00073 (0.00004)	0.00073 (0.00005)	**<0.0001**	22.57	**HS-BSP *p* < 0.0001**
						**HS-CD *p* < 0.0001**
						BSP-CD ns
PALLIDUM—S1 Mean (SD)	0.00070 (0.00002)	0.00076 (0.00004)	0.00074 (0.0004)	**<0.0001**	16.57	**HS-BSP *p* = 0.014**
						**HS-CD *p* < 0.0001**
						BSP-CD ns
PALLIDUM—SMA Mean (SD)	0.00068 (0.00003)	0.00075 (0.00003)	0.00075 (0.00005)	**<0.0001**	28,84	**HS-BSP *p* < 0.0001**
						**HS-CD *p* < 0.0001**
						BSP-CD ns
PUTAMEN—M1 Mean (SD)	0.00070 (0.00003)	0.00075 (0.00004)	0.00076 (0.00005)	**<0.0001**	17.53	**HS-BSP *p* = 0.001**
						**HS-CD *p* = 0.001**
						BSP-CD ns
PUTAMEN—S1 Mean (SD)	0.00071 (0.00002)	0.00078 (0.00005)	0.00076 (0.00005)	**<0.0001**	18.97	**HS-BSP *p* = 0.02**
						**HS-CD *p* < 0.0001**
						BSP-CD ns
PUTAMEN—SMA Mean (SD)	0.00070 (0.00003)	0.00076 (0.00004)	0.00076 (0.00005)	**<0.0001**	24.71	**HS-BSP *p* < 0.0001**
						**HS-CD *p* *< 0.0001***
						BSP-CD ns
THALAMUS—M1 Mean (SD)	0.00069 (0.00002)	0.00073 (0.00003)	0.00074 (0.00002)	**<0.0001**	21.33	**HS-BSP *p* < 0.0001**
						**HS-CD *p* < 0.0001**
						BSP-CD ns
THALAMUS—S1 Mean (SD)	0.00071 (0.00002)	0.00078 (0.00005)	0.00076 (0.00005)	**<0.0001**	18.25	**HS-BSP *p* = 0.006**
						**HS-CD *p* < 0.0001**
						BSP-CD ns
THALAMUS—SMA Mean (SD)	0.00069 (0.00002)	0.00075 (0.00003)	0.00075 (0.00005)	**<0.0001**	27.91	**HS-BSP *p* < 0.0001**
						**HS-CD *p* < 0.0001**
						BSP-CD ns

Differences between the three groups (HS, CD and BSP) were assessed using the Kruskal–Wallis test (Bonferroni corrected for multiple comparisons). H indicates the mean rank. M1 = bilateral primary motor cortex, head/face region; S1 = bilateral primary sensory cortex, head/upper limb region; SMA = supplementary motor area; SD = standard deviation. Significant *p* are shown in bold. ns = not significant.

**Table 4 biomedicines-11-00753-t004:** Axial diffusivity of subcortical-cortical WM tracts.

Axial Diffusivity						
	HS (*n* = 17)	CD (*n* = 17)	BSP (*n* = 18)	*p*	H	Post Hoc
PALLIDUM—M1 Mean (SD)	0.001065 (0.000021)	0.001116 (0.000033)	0.001117 (0.000051)	**<0.0001**	21.12	**HS-BSP *p* < 0.0001**
						**HS-CD *p* < 0.0001**
						BSP-CD ns
PALLIDUM—S1 Mean (SD)	0.001088 (0.000031)	0.001153 (0.000034)	0.001141 (0.000045)	**<0.0001**	21.14	**HS-BSP *p* = 0.002**
						**HS-CD *p* < 0.0001**
						BSP-CD ns
PALLIDUM—SMA Mean (SD)	0.001080 (0.000040)	0.001153 (0.000028)	0.001146 (0.000046)	**<0.0001**	23.91	**HS-BSP *p* = 0.001**
						**HS-CD *p* < 0.0001**
						BSP-CD ns
PUTAMEN—M1 Mean (SD)	0.001050 (0.000022)	0.001101 (0.000037)	0.001107 (0.000056)	**<0.0001**	16.07	**HS-BSP *p* = 0.002**
						**HS-CD *p* = 0.001**
						BSP-CD ns
PUTAMEN—S1 Mean (SD)	0.001064 (0.000024)	0.001137 (0.000040)	0.001124 (0.000049)	**<0.0001**	25.75	**HS-BSP *p* < 0.0001**
						**HS-CD *p* < 0.0001**
						BSP-CD ns
PUTAMEN—SMA Mean (SD)	0.001057 (0.000040)	0.001132 (0.000032)	0.001124 (0.000050)	**<0.0001**	23.58	**HS-BSP *p* = 0.001**
						**HS-CD *p* < 0.0001**
						BSP-CD ns
THALAMUS—M1 Mean (SD)	0.001068 (0.000024)	0.001122 (0.000028)	0.001124 (0.000046)	**<0.0001**	23.93	**HS-BSP *p* < 0.0001**
						**HS-CD *p* < 0.0001**
						BSP-CD ns
THALAMUS—S1 Mean (SD)	0.001113 (0.000035)	0.001186 (0.000042)	0.001166 (0.000039)	**<0.0001**	24.46	**HS-BSP *p* = 0.001**
						**HS-CD *p* < 0.0001**
						BSP-CD ns
THALAMUS—SMA Mean (SD)	0.001071 (0.000035)	0.001141 (0.000046)	0.001132 (0.000042)	**<0.0001**	26.74	**HS-BSP *p* < 0.0001**
						**HS-CD *p* < 0.0001**
						BSP-CD ns

Differences between the three groups (HS, CD and BSP) were assessed using the Kruskal–Wallis test (Bonferroni corrected for multiple comparisons). H indicates the mean rank. M1 = bilateral primary motor cortex, head/face region; S1 = bilateral primary sensory cortex, head/upper limb region; SMA = supplementary motor area; SD = standard deviation. Significant *p* are shown in bold. ns = not significant.

**Table 5 biomedicines-11-00753-t005:** Radial diffusivity of subcortical-cortical WM tracts.

Radial Diffusivity						
	HS (*n* = 17)	CD (*n* = 17)	BSP (*n* = 18)	*p* (Bonferroni)	H	Post Hoc
PALLIDUM—M1 Mean (SD)	0.000481 (0.000029)	0.000536 (0.000044)	0.000533 (0.000050)	**<0.0001**	15.83	**HS-BSP *p* = 0.005**
						**HS-CD *p* = 0.001**
						BSP-CD ns
PALLIDUM—S1 Mean (SD)	0.000505 (0.000032)	0.000559 (0.000050)	0.000539 (0.000051)	**0.005**	10.54	HS-BSP ns
						**HS-CD *p* = 0.005**
						BSP-CD ns
PALLIDUM—SMA Mean (SD)	0.000478 (0.000028)	0.000546 (0.000039)	0.000545 (0.000058)	**<0.0001**	23.73	**HS-BSP *p* < 0.0001**
						**HS-CD *p* < 0.0001**
						BSP-CD ns
PUTAMEN—M1 Mean (SD)	0.000531 (0.000034)	0.000581 (0.000047)	0.000583 (0.000054)	**0.002**	11.99	**HS-BSP *p* = 0.009**
						**HS-CD *p* = 0.007**
						BSP-CD ns
PUTAMEN—S1 Mean (SD)	0.000539 (0.000033)	0.000599 (0.000057)	0.000585 (0.000055)	**0.003**	11.32	**HS-BSP *p* = 0.041**
						**HS-CD *p* = 0.004**
						BSP-CD ns
PUTAMEN—SMA Mean (SD)	0.000515 (0.000027)	0.000577 (0.000042)	0.000572 (0.000058)	**<0.0001**	18.34	**HS-BSP *p* = 0.003**
						**HS-CD *p* < 0.0001**
						BSP-CD ns
THALAMUS—M1 Mean (SD)	0.000494 (0.000029)	0.000540 (0.000040)	0.000548 (0.000056)	**0.001**	14.0	**HS-BSP *p* = 0.005**
						**HS-CD *p* = 0.003**
						BSP-CD ns
THALAMUS—S1 Mean (SD)	0.000509 (0.000031)	0.000573 (0.000061)	0.000555 (0.000062)	**0.005**	10.78	HS-BSP ns
						**HS-CD *p* = 0.005**
						BSP-CD ns
THALAMUS—SMA Mean (SD)	0.000494 (0.000026)	0.000556 (0.000036)	0.000552 (0.000052)	**<0.0001**	21.75	**HS-BSP *p* = 0.001**
						**HS-CD *p* < 0.0001**
						BSP-CD ns

Differences between the three groups (HS, CD and BSP) were assessed using the Kruskal–Wallis test (Bonferroni corrected for multiple comparisons). H indicates the mean rank. M1 = bilateral primary motor cortex, head/face region; S1 = bilateral primary sensory cortex, head/upper limb region; SMA = supplementary motor area; SD = standard deviation. Significant *p* are shown in bold. ns = not significant.

## Data Availability

The data presented in this study are available on request from the corresponding author. The data are not publicly available due to privacy reasons.

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
