# Peer review of "Cortico-Subcortical White Matter Bundle Changes in Cervical Dystonia and Blepharospasm"

_biomedicines, 2023, doi:10.3390/biomedicines11030753_

Round 1

Reviewer 1 Report

I would like to congratulate Giannì and collaborators on their interesting work. The authors have conducted an MRI based study on the underlying white matter pathology of patients with focal dystonia. The introduction is clear, the methods described in detail.

The results are interesting in both CD and BSP patients sharing the same changes in cortico-subcortical white matter tracts, as opposed to HControls.  Using the same methodology to study other circuits in this cohort (as basal ganglia – brainstem or cerebellum) would be interesting for a future project.

I would just have a minor suggestion: the expression "whitte matter alterations" doesn't sound good. Perhaps witte matter changes, or another synonym

Studies as this one are much needed into a deeper understanding of the underlying pathogenesis of focal dystonia.

Author Response

See the file attached.

Reviewer 2 Report

Review of manuscript “Cortico-subcortical white matter bundles alterations in cervical dystonia and blepharospasm. The authors have study white matter tracts in patients which focal dystonia; blepharospasm and cervical dystonia. They found microstructural changes in both forms of dystonia compared to healthy controls. The study is well-performed and could be useful in development of clinical scales in patients with focal dystonia. The images included are illustrative. I have some smaller comments.

Methods

Did you include adults only or also pediatric patients?

Were any of the patients under some sort of treatment?

It seems like the terms healthy subject and healthy control are used alternately, perhaps only use one?

Author Response

See the file attached.
